# Differential Recruitment of DNA Repair Proteins KU70/80 and RAD51 upon Microbeam Irradiation with α-Particles

**DOI:** 10.3390/biology11111652

**Published:** 2022-11-11

**Authors:** Laure Bobyk, François Vianna, Juan S. Martinez, Gaëtan Gruel, Marc Benderitter, Céline Baldeyron

**Affiliations:** 1Institut de Radioprotection et de Sûreté Nucléaire (IRSN), PSE-SANTE, Service de Recherche en Radiobiologie et en Médecine Régénérative (SERAMED), Laboratoire de Radiobiologie des Expositions Accidentelles (LRAcc), F-92262 Fontenay aux Roses, France; 2Institut de Radioprotection et de Sûreté Nucléaire (IRSN), PSE-SANTE, Service de Recherches en Dosimétrie (SDOS), Laboratoire de Micro-Irradiation, de Métrologie et de Dosimétrie des Neutrons (LMDN), F-13115 Cadarache, France; 3Institut de Radioprotection et de Sûreté Nucléaire (IRSN), PSE-SANTE, F-92262 Fontenay aux Roses, France

**Keywords:** microbeam, α-particle, double strand break, DNA repair

## Abstract

**Simple Summary:**

Human populations can be exposed to ionizing radiation (IR) in different circumstances (natural, medical, and industrial). By interacting with the matter, IR causes damage in all cellular compartments, including DNA. Mammalian cells possess several different DNA repair mechanisms for removing IR-induced DNA damage among which DNA double-strand breaks (DSBs) are the most deleterious lesions. In this work, we investigated the temporal dynamics of the two major DSB repair pathways upon α-particle irradiation delivered by the MIRCOM microbeam. We found that these mechanisms are differentially recruited at IR-induced DNA damage sites.

**Abstract:**

In addition to representing a significant part of the natural background radiation exposure, α-particles are thought to be a powerful tool for targeted radiotherapy treatments. Understanding the molecular mechanisms of recognition, signaling, and repair of α-particle-induced DNA damage is not only important in assessing the risk associated with human exposure, but can also potentially help in identifying ways of improving the efficacy of radiation treatment. α-particles (He^2+^ ions), as well as other types of ionizing radiation, and can cause a wide variety of DNA lesions, including DNA double-strand breaks (DSBs). In mammalian cells, DNA DSBs can be repaired by two major pathways: non-homologous end-joining (NHEJ) and homologous recombination (HR). Here, we investigated their dynamics in mouse NIH-3T3 cells through the recruitment of key proteins, such as the KU heterodimer for NHEJ and RAD51 for HR upon localized α-particle irradiation. To deliver α-particles, we used the MIRCOM microbeam, which allows targeting of subnuclear structures with submicron accuracy. Using mouse NIH-3T3 cells, we found that the KU heterodimer is recruited much earlier at DNA damage sites marked by H2AX phosphorylation than RAD51. We also observed that the difference in the response of the KU complex and RAD51 is not only in terms of time, but also in function of the chromatin nature. The use of a microbeam such as MIRCOM, represents a powerful tool to study more precisely the cellular response to ionizing irradiation in a spatiotemporal fashion at the molecular level.

## 1. Introduction

Human populations are daily exposed to ionizing radiation (IR) from different sources: natural, medical, and industrial. Medical and industrial exposure concerns mostly low linear energy transfer (LET) radiations (X and γ radiations), which have been widely studied. A significant part of the natural background radiation exposure results from α-particles (He^2+^ ions) due to the inhalation of radon gas [1]. Indeed, radon is a naturally occurring radionuclide in the environment, which during decay emits high LET α-particles. Since radon has been linked with the onset of lung cancer when inhaled over many years, it is considered a health concern. Radon-222 has been classified by the International Agency for Research on Cancer (IARC) as being carcinogenic to humans [2]. Lastly, a smaller part of α-particles exposure comes from the targeted radiation therapy where α-emitting radionuclides are specifically localized to deliver a cytotoxic radiation dose to cancerous tissue, while sparing surrounding healthy tissues [3].

If high stopping power of α-particles makes external exposure relatively safe, internal exposure induces cell lesions, among them the damage of the DNA macromolecule. In fact, as all ionizing radiations, α-particles induce DNA double-strand breaks (DSBs), which are amongst the most deleterious lesions since they may lead to genomic instability and even cell death. In recent years, laser microbeams generating different types of DNA damage have played a major role in the study of the temporal and spatial organization of the cellular DNA damage response [4]. These approaches allow the induction of DNA damage in a defined region in the cell nucleus in situ with a micrometric precision and permits the monitoring of recruitment kinetics of DNA damage response (DDR) proteins to localized DNA damage sites, as well as the surrounding chromatin organization modifications [5]. However, the characteristics and complexity of the lesions generated by the different laser systems have not been clearly determined and are undeniably different from those induced by ionizing radiation [6,7]. The ion microbeam technology thus offers the possibility to deliver a predetermined number of particles of a certain radiation quality (type and energy) in a specific area within a cell, nucleus, or cytoplasm, with micrometric spatial resolution [8].

Mammalian cells possess several different DNA repair mechanisms for removing IR-induced DNA DSBs. Among these mechanisms, canonical non-homologous end-joining (NHEJ) and homologous recombination (HR) are the two major pathways to recover DSBs, while single-strand annealing (SSA) and alternative NHEJ (alt-NHEJ, sometimes called microhomology mediated NHEJ) can repair the residual DSBs that are unable to be repaired by NHEJ and HR [9,10]. The HR pathway, requiring the presence of a homologous sequence on the sister chromatid to guide the repair, only occurs in late S and G2 phases, whereas NHEJ consisting in the joining of DSB ends, operates at all stages of the cell cycle [11,12]. Like HR, SSA is active during the late S and G2 phases and is also a homology-dependent repair mechanism, but it is error-prone [13]. The alt-NHEJ pathway is a DNA end resection-dependent mechanism that uses microhomologies near the break for repair and although active throughout the cell cycle, alt-NHEJ shows maximum activity in G2 [10,14]. Although the way to repair a DSB is dependent on the cell cycle, the chromatin organization surrounding the DSB significantly affects the kinetics of DNA repair. The subsequent DNA damage response causes significant changes to the chromatin environment at the DSB [11,15], such as the phosphorylation of the histone variant H2AX on serine 139, known as γH2AX, spreading over megabases of the adjacent chromatin [16,17].

In this work, we investigated the temporal dynamics of the two major DSB repair pathways, which can be undoubtedly identified through certain of their protein players. For this, we used mouse NIH-3T3 cells, a mammalian cellular model that is widely used for the study of DNA repair and the surrounding chromatin architecture dynamics [18,19,20]. We generated DNA damage in these cells with α-particle irradiation delivered by the MIRCOM microbeam facility (IRSN, Cadarache, France), which is dedicated to radiobiology experiments and which thanks to its micrometer precision targeting allows the exclusive irradiation of specific areas within cell nuclei [21]. Following irradiation with α-particles, we evaluated the recruitment kinetics of the KU70/80 heterodimer (KU), a DSB sensor in NHEJ and RAD51, the recombinase involved in the formation of the nucleoprotein filament that drives HR-mediated DSB repair. The well-characterized marker of DNA DSBs, γH2AX allowed the visualization of DSBs at sites of irradiation. We also studied the cell cycle of NIH-3T3 cells over time following irradiation with α-particles and we analyzed the localization of γH2AX, KU and RAD51 at the DNA damage sites in function of the type of chromatin structure upon irradiation.

## 2. Materials and Methods

### 2.1. Cell Culture

We obtained NIH-3T3 cells (CRL-1658) from the American Type Culture Collection (ATCC, LGC Standards, Molsheim, France). We maintained these cells in DMEM-GlutaMax medium (Gibco, Life Technologies, France) supplemented with 10% iron fortified calf bovine serum (ATCC, LGC Standards, Molsheim, France), 100 U/mL penicillin and 100 μg/mL streptomycin (Gibco, Life Technologies, Villebon-sur-Yvette, France) in a humidified atmosphere with 5% CO_2_ at 37 °C.

For irradiation, we seeded asynchronous cells at least 24 h beforehand on 4 μm-thick polypropylene foil (Goodfellow, Lille, France) coated with Cell-Tak (2 μg/cm^2^, Corning, BD Biosciences, Le Pont de Claix, France) on specific PEEK (Polyether ether ketone) dishes, as described in Bourret et al. [22] in order to obtain about 10,000 cells/cm^2^ at the moment of irradiation. After plating, we added nutrient medium and kept cells in a humidified atmosphere with 5% CO_2_ at 37 °C until irradiation. Just before irradiation, we stained cells with 150 nM Hoechst dye 33,342 (Sigma-Aldrich, St. Quentin Fallavier, France) in culture medium for 30 min to 1 h and we reincubated cells in fresh medium after a PBS wash.

### 2.2. Microbeam Irradiation

We performed the irradiation of samples by using the microbeam of the MIRCOM (“Microfaisceau d’Ions pour la Radiobiologie aux échelles des Cellules et Organismes Multicellulaires”—ion microbeam for radiation biology at the cellular and multicellular scales) facility [21], operated by the Institut de Radioprotection et de Sûreté Nucléaire (IRSN) in Cadarache, France.

#### 2.2.1. Beamline

This facility is equipped with a 2 MV Tandetron™ accelerator manufactured by High Voltage Engineering Europa B.V. (HVEE, Amersfoort, The Netherlands), and originally used to produce reference monoenergetic neutron fields [23]. It can produce different focused ions beams: protons up to 4 MeV, α particles up to 6 MeV, and carbon or heavier ions up to 8 to 12 MeV, depending on the charge state of these ions. Downstream of the accelerator is the microbeam line itself. For more details, the whole beamline and the working principle of MIRCOM were detailed by Vianna et al. [21]. Briefly, after collimation and focusing with four magnetic quadrupoles in a “Russian quadruplet” configuration, the microbeam is extracted in air through a 150 nm thick Si_3_N_4_ extraction window (Silson Ltd., Southampton, UK) and goes through a 250 μm-thick residual layer of air before reaching the cell dish placed in a specific holder. An inverted epifluorescence microscope (AxioObserver™ Z1, Carl Zeiss Microscopy GmbH, Jena, Germany) equipped with a 37 °C heating chamber is positioned on the other side of the cell dish to select areas of interest in the culture dish, and to choose the relevant targets to irradiate, either by manual selection, or by automatic shape recognition. The microbeam is then sent on target by electrostatic scanning plates, either for a given number of ions or for a given time.

#### 2.2.2. Dose Control

We irradiated cells with a mean number of ions, controlled by the definition of an opening time of the microbeam on each target, as described previously [21]. To monitor the beam and thus define this opening time, we used a PIPS detector (PD50-12-100AM, Mirion Technologies (Canberra) S.A.S., Montigny-Le-Bretonneux, France), located on the microscope objective wheel. This detector can easily be placed in the beam trajectory, but can only be used without any living sample, because of the limited range of ions available on MIRCOM inside the samples. It is used as a reference detector to monitor the counting rate of the ion beam. An automatic procedure allows the multiplication of single measurements to ensure a reliable result. An example of the measurements carried out during the experimental campaigns, for an opening time of 100 ms, is shown in Figure 1a. In this particular case, we obtained a mean number of ions N = 150, with a standard deviation σ = 12. This result is compatible with a Poisson law, where σ = √N. To ensure that the beam did not undergo major fluctuations during the irradiation, we measured the mean counting rate between each sample. We observed no significant variation of the beam between two consecutive controls.

#### 2.2.3. Irradiation Parameters

We targeted cells at the center of their nucleus and irradiated them with a pattern of 9 spots spread across an 8 μm-wide horizontal line (Figure 1b). We carried out irradiation in an easily recognizable damage pattern, that would be distinguishable from spontaneous damage. We delivered a mean number of 50 ± 7 α-particles on each spot, resulting in a relative fluctuation of 14%. We fixed this quantity of α-particles because it is the minimum amount allowing the detection of RAD51 recruitment under our experimental conditions. In this configuration, the mean irradiation speed was approximately 300 ms. The LET of the α-particles after going through the extraction window (150 nm), the residual air layer (250 μm), and the polypropylene foil (4 μm) was 84 keV/μm, as was calculated using SRIM-2013 [24]. Thus, the mean specific energy was approximately 5.0 ± 0.7 Gy per spot. Due to limited beam time availability, the number of irradiated samples was restricted.

### 2.3. Immunostaining and Confocal Microscopy

After irradiation, we reincubated cells at 37 °C in fresh medium. At indicated times after irradiation, we washed the cells twice in PBS (Gibco, Life Technologies, Villebon-sur-Yvette, France) and we performed in situ cell fractionation as described previously [25]. Briefly, we rinsed cells with cytoskeleton buffer (CSK) containing 10 mM PIPES pH 6.8, 100 mM NaCl, 300 mM sucrose, 3 mM MgCl_2_ (Sigma-Aldrich, France), and a cocktail of protease inhibitors (Complete, EDTA-free tablets, Roche, MERCK, Saint-Quentin-Fallavier, France). We subsequently performed a Triton X-100 extraction by incubating cells in CSK containing 0.5% Triton X-100 (Sigma-Aldrich, Saint-Quentin-Fallavier, France) for 5 min on ice. After washes with CSK and PBS, we fixed cells with 2% wt/vol paraformaldehyde (EMS, Euromedex, Souffelweyersheim, France) for 20 min at room temperature (RT). After one wash with PBS and two washes with PBT (PBS containing 0.1% vol/vol Tween 20 (Sigma-Aldrich), we incubated cells in 5% BSA (Sigma-Aldrich, Saint-Quentin-Fallavier, France) in PBT for 5 min at RT and subsequently with the appropriate primary antibodies diluted in blocking buffer. Since we did not find commercial antibodies against KU and RAD51 proteins allowing co-staining, we used two mixtures of primary antibodies: (1) mouse anti-γH2AX (Millipore, St Quentin en Yvelines, France, 05-636; 1:2000) and rabbit anti-KU70/80 (Abcam, Cambridge, UK, ab 53126; 1:400) antibodies for 1 h at RT, or (2) mouse anti-γH2AX (Millipore, St Quentin en Yvelines, France, 05-636, 1:2000) and rabbit anti-RAD51 (Abcam, Cambridge, UK, ab137323; 1:400) antibodies for overnight incubation at 4 °C. After three washes with PBS and one with blocking buffer, we incubated cells with secondary antibodies: Alexa Fluor 488 donkey anti-mouse and Alexa Fluor 594 donkey anti-rabbit (ThermoFisher, Illkirch, France, A21202 and A21207, respectively; 1:1000) for 1 h at RT. We washed them three times in PBT, and incubated them with 0.5 μg/mL DAPI (Molecular Probes, Life Technologies, Villebon-sur-Yvette, France) in PBS for 5 min. After three washes in PBS, we mounted samples onto slides with ProLong Diamond mounting medium (Molecular Probes, Life Technologies, Villebon-sur-Yvette, France).

We recorded images on a right confocal LSM780NLO microscope (Carl Zeiss MicroImaging GmbH, Jena, Germany), equipped with a Plan-Apochromat 63×/1.4 NA oil M27 objective and piloted with the Zen Black 2011 SP4 software. We analyzed serial z-stack images by using ImageJ 1.47v software (Rasband, W.S., ImageJ, U.S. National Institutes of Health, Bethesda, MD, USA, https://imagej.nih.gov/ij/, 1997–2018). For quantitative analysis, we performed a maximum intensity projection to display data in a single plane image. We called positive cells, those showing a specific staining for the proteins of interest, corresponding to the irradiation pattern drawn, 9 spots spread across an 8 μm wide horizontal line (Figure 1b). We visually scored γH2AX-positive cells showing KU70/80 or RAD51 staining. We analyzed a minimum number of 150 nuclei for each time-point.

### 2.4. Quantitative Image-Based Cell Cycle Analysis by Epifluorescence Microscopy

In order to determine how irradiated cells progress through the cell cycle, we reacquired the same samples as above (see “Immunostaining and confocal microscopy” paragraph) and analyzed images on a Scan^R platform (Olympus, Rungis, France), as described previously [26,27]. We recorded images on an inverted epifluorescence Olympus IX81 microscope with a 10×/0.25 NA objective; the microscope was coupled with a motorized SCAN IM IX2 stage (Märzhäuser, Wetzlar, Germany), a MT20 fluorescence illumination system with a fast filter wheel and an Orca R2 CCD camera (Hamamatsu, Massy, France). We adjusted the acquisition times for the different channels to obtain images under non-saturating conditions (i.e., in the 12-bit dynamic range) for all the experimental points analyzed. After acquisition, we performed image analysis with the ScanˆR analysis software (Olympus, Rungis, France). We used an edge segmentation algorithm implemented in the software, which is based on Canny’s method to detect nuclei in the DAPI channel (main object) and γH2AX foci in the FITC channel (sub-object). We performed a first selection based on the area and circularity of the nuclei to consider only isolated nuclei and to remove from the analysis objects corresponding to clusters of nuclei and cellular debris. As in a flow-cytometry analysis, the cells were distributed in different phases of the cell cycle by assessing the integrated intensity of the DAPI signal (DNA content) within the entire nucleus. We visually determined the region containing the irradiated cells through the γH2AX signal, corresponding to the drawn pattern of a line of 9 spots in irradiated nuclei. Finally, we determined the cell cycle stage of all irradiated cells and γH2AX-positive cells. We analyzed a minimum number of 80 nuclei per experiment.

### 2.5. Statistical Analysis

The data shown result from at least two independent experiments. In all experiments, we compared the time-point 5 min to the other time-points, and we calculated *p*-values using Student’s *t* test.

## 3. Results

### 3.1. Cell Cycle Progression and Cell Mortality of NIH-3T3 Cells upon α-Particle Irradiation

As repair mechanisms depend on the cell cycle [15], we determined the distribution of NIH-3T3 cells in the cell cycle based on their DNA content by measuring the integrated intensity of DAPI within the cell nuclei [26,28] before and after irradiation with α-particles. The distribution of cells in the different phases of the cell cycle at the moment of irradiation is approximately 58 ± 4% of cells in G1, 21 ± 3% in S and 15 ± 2% in G2 (Figure 2a, 5 min). This initial distribution among the phases of the cell cycle gives the probability of causing DNA damage in G1, S, or G2 in cells randomly chosen within the cell monolayer. We irradiated a mean of 38 ± 18 NIH-3T3 cell nuclei per ×20 microscopy field with a pattern of one line of 9 dots spaced by 1 μm each with 50 α-particles per dot (Figure 1b). Irradiation was carried out in a clearly recognizable damage pattern (line) in order to easily distinguish the induced damage from background. We found that no significant mortality is observed in the 24 h post-exposure, as we recovered all the irradiated nuclei within the targeting area for all given post-irradiation times.

We found that 5 min after irradiation, the targeted cells show the same cell cycle distribution as neighboring unirradiated cells (data not shown), arguing that in 5 min no obvious change in the cell cycle distribution is observed. Six hours upon irradiation, we observed a slight increase (about 25%) of cells in G2 (Figure 2a) indicating a radio-induced G2 arrest, still observable 24 h after exposure. Focusing on the cells having a remaining γH2AX-positive signal 24 h post-irradiation (Figure 2b), we noticed that they are mostly in the G2 phase of the cell cycle, suggesting that unrepaired cells 24 h upon exposure are predominantly blocked in G2 and should mostly correspond to those initially damaged in S or G2.

Our data are in agreement with the literature [29,30] since we showed that α-particle irradiation clearly delays the progression through the cell cycle with a major arrest in G2 (Figure 2).

### 3.2. Kinetics of Signaling and Repair of DNA DSBs after α-Particle Irradiation

In order to improve our knowledge on the consequences of irradiation with α-particles on DNA, we determined the kinetics of H2AX phosphorylation on ser139 (γH2AX), which is one of the earlier steps in DDR [12], and of the recruitment of the KU complex and RAD51 proteins, which are involved in the two main DSB repair mechanisms, NHEJ and HR, respectively. We used the same samples from the cell cycle experiments (see above).

Firstly, 5 min after irradiation, almost 100% of irradiated cells exhibit clear γH2AX staining (Figure 3) along a signal shape that is consistent with the irradiation pattern (Figure 1b and Figure 4a,b). Next, we evaluated the localization of KU70/80 dimer or RAD51 protein at irradiated sites, on which unsurprisingly γH2AX is detected. We observed the recruitment of KU70/80 dimers at the damage sites shortly (5 min) after irradiation in all cells (Figure 4a,c, 5 min). In contrast, although around 35% of irradiated NIH-3T3 cells are in S or G2 phase (Figure 2a), a period when HR-mediated repair is active, the recruitment of RAD51 at irradiated sites is not observed in any of the cells 5 min upon localized irradiation (Figure 4b,c, 5 min). These data suggest a faster recruitment of NHEJ proteins on damaged DNA in comparison to HR proteins following an irradiation with α-particles.

One hour post-irradiation, the γH2AX signal becomes systematically, but transiently, pan-nuclear or perinuclear in all targeted cells (Figure 4a,b, 1 h). At this time-point of 1 h, even if the γH2AX signalization seems to disappear from the DNA damage sites, the presence of the KU complex is still observable in accordance with the irradiation pattern in 98% of the irradiated cell nuclei (Figure 4a, 1 h). The recruitment of RAD51 to DNA damage sites is now detectable, but in a smaller number of irradiated cells (9% approx.), even though ~40% of irradiated cells are in S or G2 phase (Figure 2a). The persistence of these DNA repair proteins, KU dimers and RAD51, at the damage induction sites makes unlikely the hypothesis of a general chromatin remodeling explaining the spreading of γH2AX to the whole nucleus. Moreover, 6 h and 24 h post-irradiation, in cells with a remaining γH2AX signal (respectively 70% and 50% of initially irradiated cells, Figure 3), γH2AX localization is again observed at the irradiation-induced damage sites (Figure 4a,b, 6 h and 24 h).

We noticed that the γH2AX signal decreases in intensity over time (data not shown) and that the γH2AX pattern significantly becomes punctuated (Figure 4a,b). Finally, 24 h after exposure, around 50% of irradiated cells do not have detectable γH2AX signal anymore and are thus likely to have repaired their irradiation-induced DNA damage (Figure 3). The decrease of γH2AX-positive cells appears to be correlated with the decrease of the KU heterodimer at the DNA damage sites (Figure 3 and Figure 4c). On the other hand, the proportion of γH2AX-positive cells with RAD51 signal increases from 30% at 6 h to about 50% at 24 h (Figure 4c), even though 85% of γH2AX-positive cells at 24h post-irradiation are in S/G2 (Figure 2b). Together, these data suggest that HR-mediated repair in NIH-3T3 cells occurs with slower kinetics following intensive α-particle irradiation.

In conclusion, our data suggest that NHEJ proteins are recruited much faster on damaged DNA sites than HR proteins following an intensive local α-particle irradiation. Moreover, our results show that at times when γH2AX signal is disappearing (6 h and 24 h upon exposure), the percentage of γH2AX-positive cells with a KU signal decreases, while the percentage of γH2AX-positive cells with a RAD51 signal seems to still increase, suggesting a differential processing of DSBs by NHEJ and HR after a localized α-particle irradiation.

### 3.3. Recruitment of DNA Repair Proteins to Constitutive Heterochromatin

Since chromatin organization also participates in the regulation of the balance between the different DNA repair pathways [15,31], we decided to evaluate the recruitment of DNA repair proteins to constitutive heterochromatin. The NIH-3T3 cell line is a good model for this kind of investigation, because its constitutive heterochromatin can be readily visualized as chromocenters, which stain densely with fluorescent DNA dyes [19]. We found that 5 min after α-particle irradiation, the KU70/80 heterodimer and γH2AX are observed inside and outside chromocenters (Figure 5a). As described above, RAD51 labelling is never observed at this time point. However, at 24 h post-irradiation when the RAD51 signaling becomes significant, RAD51, γH2AX, and even remaining KU70/80 were systematically absent from the irradiated chromocenters, but present at their periphery (Figure 5a,b).

In agreement with the literature [32], we observed that soon after localized α-particle irradiation, γH2AX is found inside the chromocenter before its relocation to the chromocenter periphery at a later time point after irradiation. As previously shown for XRCC1 [32], upon α-particle irradiation the KU complex is detected within the chromocenter at an early time-point and then at the borders of the chromocenter at later time-points. In line with the fact that HR is inhibited within heterochromatin compartments [33], we saw that RAD51 is never located inside the damaged chromocenters upon irradiation, but is always in its periphery.

## 4. Discussion

Exposure to α-particles both by radon emission from the soil and by the use of α-radionuclide-targeted therapy requires a thorough knowledge of the impact of this densely ionizing radiation on tissue. In fact, a better understanding of how DNA damage induced by α-particles is processed would allow a better evaluation of the risks for the concerned populations, as well as their optimal use in cancer therapy. While the induction of localized DNA damage by laser microirradiation has made it possible to understand many steps in DNA damage response (DDR) [4], there are undeniable differences between the nature and the structure of DNA lesions induced by laser irradiation and those induced by irradiation with α-particles [6]. Thus, we used the advantages offered by ionizing radiation microbeams such as MIRCOM, which allow precise delivery of charged particles with high spatial accuracy [6]. We analyzed the behavior of two major DSB repair pathways through the recruitment of two markers, KU70/80 for NHEJ and RAD51 for HR, on localized DNA damage sites. In addition, we paid particular attention to the cell cycle distribution of the exposed cells, which influences the probability of observing the abovementioned DNA repair pathways in action. This point will importantly allow us to put into perspective the observations concerning the kinetics of such DNA repair markers.

All the cells targeted with the α-particle microbeam present rapid phosphorylation of H2AX after irradiation. This result confirms the high precision of targeting with the MIRCOM microbeam line [21]. In addition, even though we drew a pattern of 9 dots spaced of 1 μm with the MIRCOM microbeam, the targeted cells carried γH2AX staining in a continuous line, this reflected its well-known spreading on chromatin adjacent to DNA breaks [16,17]. Here, and for the first time, we report the recruitment of KU70/80 dimer onto DNA damage sites after irradiation with α-particles by using indirect immunofluorescence. This complex is present in all irradiated cells 5 min upon irradiation regardless of the cell cycle stage, which is consistent with the rapid implementation of the NHEJ repair mechanism, and this, irrespective of cell cycle phase in mammalian cells. Our data are in agreement with previous studies based on the use of heavy ion microbeams in which the recruitment of GFP-fused KU80 and YFP-fused DNA-PKcs proteins has been observed a few seconds upon DNA damage induction in living cells [34,35]. At the 5 min time-point, even if ~35% of the NIH-3T3 cells locally irradiated with α-particles are in S/G2, the recruitment of RAD51 is not observed. This is consistent with several prior studies showing in mammalian cells that RAD51 proteins start to be observed at damage sites 15 min upon irradiation with α-particles from an Americium source [36,37,38]. In contrast to NHEJ which is a rapid mechanism, HR operates with slower kinetics, due to the fact that it is a multiple step process that additionally requires extensive chromatin changes in the DSB vicinity [15].

One hour post-irradiation, we surprisingly found that the γH2AX signal is transiently pan-nuclear (or perinuclear). This type of signal is generally interpreted as a potential sign of cell death engagement, as described in a recent review by Solier et Pommier [39]. However, our data showed that this phenomenon is transient since we detected no cell mortality during the 24 h following the exposure. A similar observation has been reported by Meyer et al. [40] after localizing ion irradiation of mammalian cells. The authors reported that the phosphorylation of H2AX in undamaged chromatin over the whole cell nucleus is likely to be related to the amount and the complexity of induced DNA lesions.

Since the processing of DSB clusters induced by densely ionizing radiation (as α-particles) might depend on cell cycle phase and chromatin state [41,42], we characterized the distribution of irradiated NIH-3T3 cells into the cell cycle at each time point studied upon irradiation. At the time of DNA damage induction almost 35% of cells were in S/G2, thus being able to implement HR. Surprisingly, this particular cell population is more likely to have a delay in DNA repair, since we observed 24 h post-exposure that more than 70% of cells still having a γH2AX signal were arrested in G2, with a great proportion of cells still carrying KU70/80 staining. Our finding would seem to illustrate the notion that NHEJ is believed to be the mechanism predominantly used to repair DSBs and that when this pathway is hindered (here probably due to the complexity of DNA damage induced by densely ionizing radiation), the repair is switched to HR, a more conservative mechanism that requires extensive chromatin modification to allow the subsequent elaborate steps, such as DSB-end resection, RAD51 loading, initiation of sister chromatid pairing and branch migration [15].

Heterochromatin, a highly compact chromatin architecture, seems to constitute a barrier for DSB repair. Recently, DSBs induced in heterochromatic domains have been proposed to be relocated outside of heterochromatin to be repaired [18]. Thus, we investigated the localization of the two DNA repair markers, KU70/80 and RAD51, with respect to chromocenters. We found that γH2AX and KU70/80 staining appears within the chromocenters of NIH-3T3 cells 5 min upon irradiation with α-particles, whereas at a later time point 24 h post-irradiation, they are present at the periphery of pericentric heterochromatin domains. In our experimental conditions, more than 70% of cells are arrested in G2 at 24 h. In a publication from 2016 where the CRISPR/Cas9 technology was used to specifically induce DSBs in heterochromatin structures [18], the authors found that while the KU complex is found in the core of pericentric heterochromatin domains in NIH-3T3 cells in G1, DSBs relocate to outside the chromocenters in S/G2 cells. In addition, our data show that in cells blocked in G2, RAD51 proteins accumulate exclusively at the periphery of chromocenters, where γH2AX is found. This peripheral RAD51 localization is thought to occur in order to avoid recombination between repetitive sequences that are often found in heterochromatin [33].

Since our results are compatible and complementary with the current knowledge of DSB repair in mammalian cells, the use of a microbeam such as MIRCOM, represents a powerful tool to study in a more precise manner the cellular response to ionizing irradiation in a spatiotemporal fashion at the molecular level.

## 5. Conclusions

Understanding how IR-induced DSBs are processed at the molecular level would allow a better assessment of risks incurred by the population, but also their optimal use in anti-tumor therapy. By using the MIRCOM microbeam, we targeted α-particles with submicron accuracy within nuclei of NIH-3T3 cells and we observed the phosphorylation of H2AX at localized DNA damage sites and subsequently the transient spreading of γH2AX signal through the whole cell nuclei, without leading to cell death. The biological parameters (type of charged particles, their number, their distribution within cell nucleus or the role of proteins that anchor on γH2AX), which drive to this γH2AX behavior are not fully characterized yet. We also found in NIH-3T3 cells that the KU heterodimer, involved in NHEJ-mediated DSB repair, is recruited early at DNA damage sites marked by H2AX phosphorylation, while RAD51, the key protein of HR, arrives to DNA damage sites at a later time, although a part of the cell population is in S/G2. Moreover, in NIH-3T3 cells the difference observed in the response of the KU complex and RAD51 is not only in terms of time, but also in localization (i.e., chromatin vs. heterochromatin). Importantly, our data highlight that analysis of DNA damage repair after IR should take into account several biological parameters, not only the response time upon damage induction and the identity of the proteins involved, but also the sub-nuclear localization (i.e., chromatin state) as well as the cell cycle stage of the cells.

## Figures and Tables

**Figure 1 biology-11-01652-f001:**
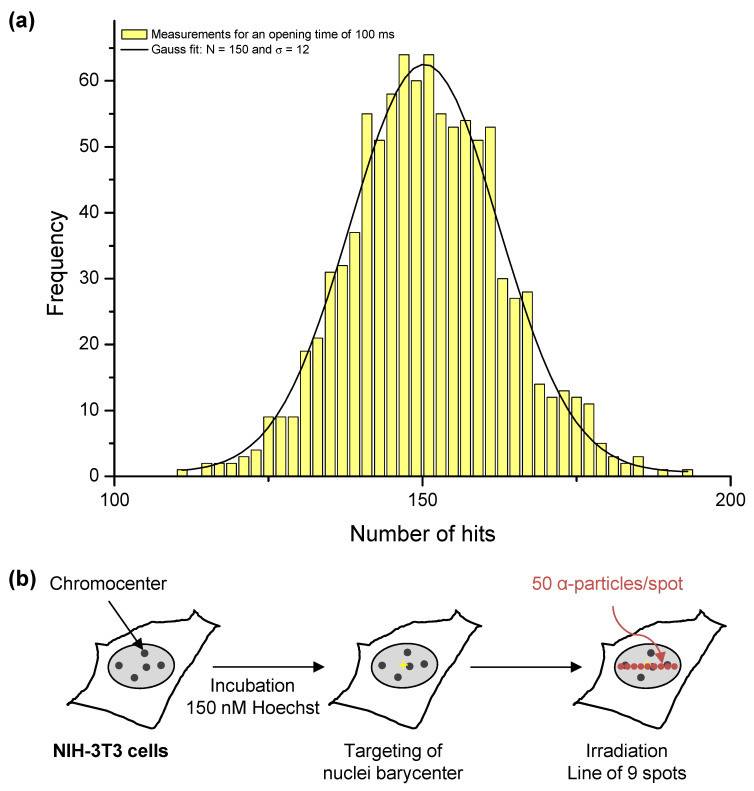
Targeted irradiation on the MIRCOM facility. (**a**) Hit number distribution, used to monitor the mean counting rate of the microbeam between each irradiation. In this example, we performed 961 measurements with a beam opening time of 100 ms. The mean number of ions is 150, with a standard deviation σ = 12, and the relative fluctuation in the delivered hits is thus 8%. (**b**) Experimental scheme for targeted irradiation in a dotted line pattern. We centered within the cell nucleus a pattern of 9 spots spread across an 8 μm-wide horizontal line and we carried out irradiation by delivering a mean number of 50 ± 7 α-particles on each spot, resulting in a relative fluctuation of 14%.

**Figure 2 biology-11-01652-f002:**
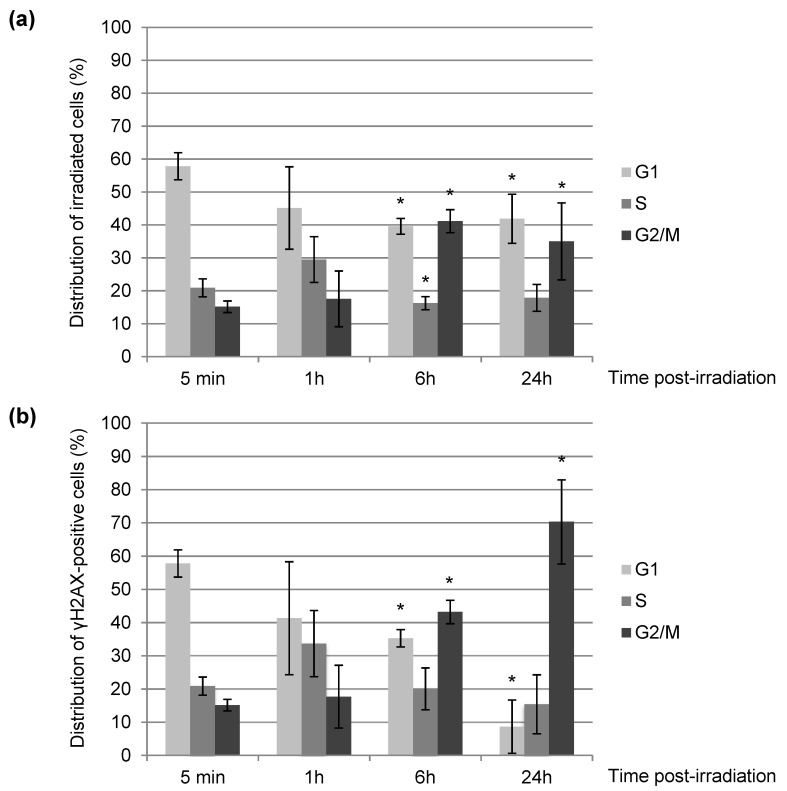
Cell cycle distribution of NIH-3T3 cells upon irradiation with α-particles. We irradiated NIH-3T3 cell nuclei with α-particles distributed in a dotted line, as shown in Figure 1b. At the indicated times after irradiation, we permeabilized cells with CSK + Triton X-100 to remove soluble nuclear components and fixed them. We subsequently performed immunostaining of γH2AX and DNA staining with DAPI. We monitored the cell cycle status (**a**) of irradiated cells and (**b**) of γH2AX-positive cells among irradiated cells by immunofluorescence. Each value shown represents the mean of at least three independent irradiations. For each value, we analyzed in total between 80 and 800 cells. The error bars represent the standard error, and the *p* values are indicated (* *p* < 0.05).

**Figure 3 biology-11-01652-f003:**
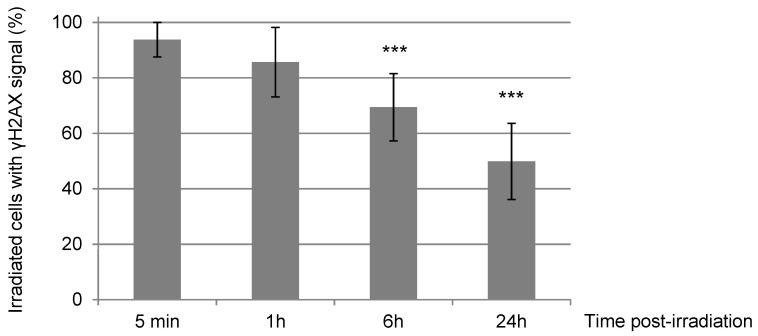
Percentage of γH2AX-positive NIH-3T3 cells upon irradiation with α-particles. We quantified the number of NIH-3T3 cells from samples used previously in Figure 2 carrying γH2AX signal among the total irradiated cells at the indicated times after α-particle irradiation. Each value shown represents the mean of at least three independent irradiations. The error bars represent the standard error, and the *p* values are indicated (*** *p* < 0.001).

**Figure 4 biology-11-01652-f004:**
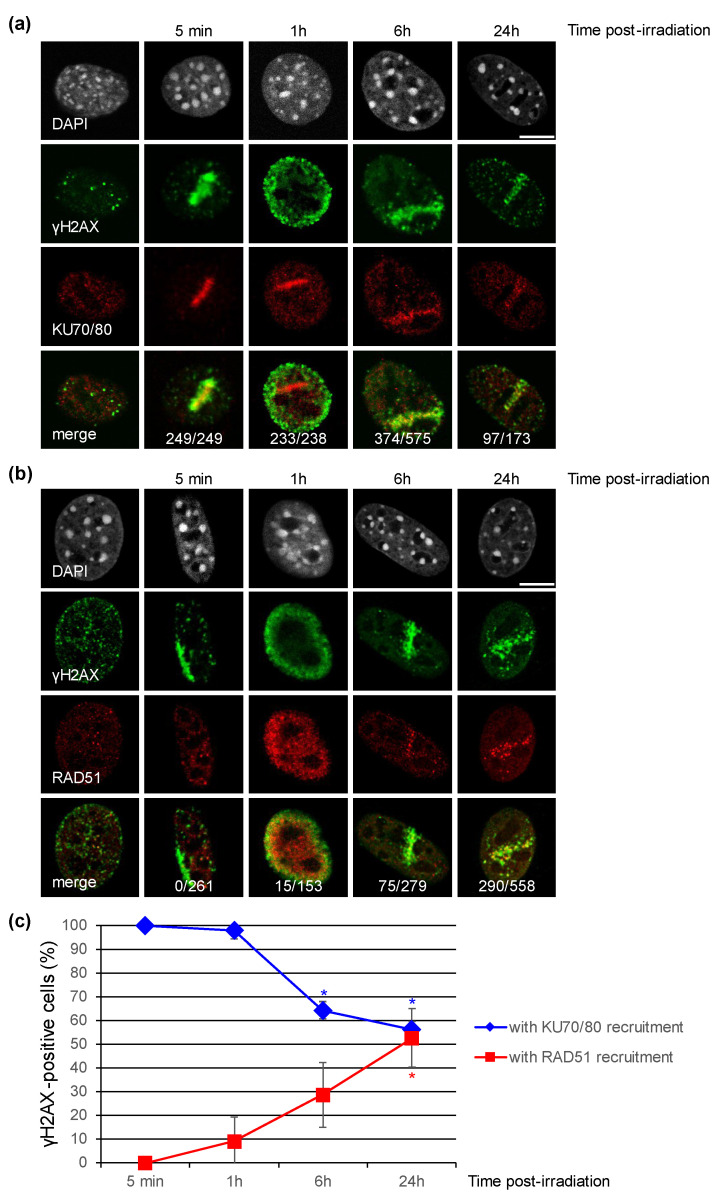
Recruitment of the KU complex or RAD51 to DNA damage sites induced by a microbeam irradiation with α-particles in NIH-3T3 cells. (**a**,**b**) Accumulation of the KU complex and RAD51 proteins to DNA damage sites detected by γH2AX staining at the indicated time upon irradiation. The samples used in Figure 2 were subjected to co-immunostaining with antibodies against γH2AX and KU70/KU80 (**a**) or RAD51 (**b**). In the non-irradiated cell panels, a representative cell displaying KU70/80 staining (**a**) and a cell in S phase with RAD51 staining (**b**) are shown. DNA was stained with DAPI. The total number of γ-H2AX-positive cells carrying KU70/80 signal or RAD51 signal is indicated for each time point. Bars represent 10 μm. (**c**) Percentage of γH2AX-positive cells with KU70/80 signal or with RAD51 signal. Each value shown represents the mean of at least two independent experiments from (**a**,**b**). Error bars represent the standard error, and the *p* values are indicated (* *p* < 0.05).

**Figure 5 biology-11-01652-f005:**
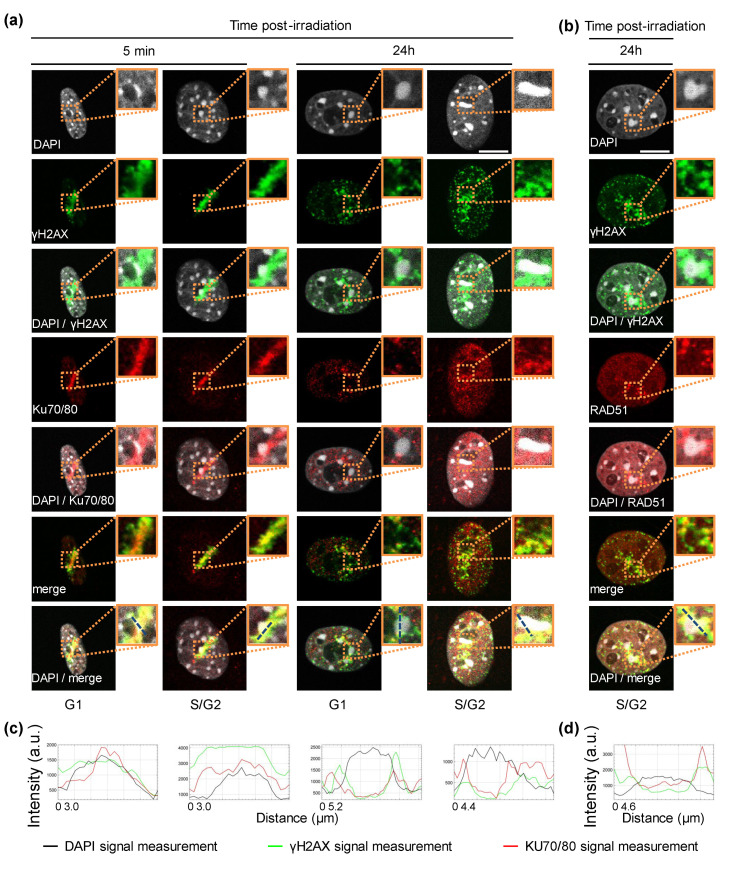
Recruitment of the KU complex and RAD51 at the chromocenters of NIH-3T3 cells irradiated with α-particles. (**a**,**b**) Localization of the KU70/80 dimer and RAD51 protein relative to chromocenters 5 min and 24 h post-irradiation. For the indicated time after localized α-particle irradiation, we re-analyzed irradiated NIH-3T3 cells, which were part of the data of Figure 4, and which were immunostained with antibodies against γH2AX and KU70/KU80 (**a**) or RAD51 (**b**). Enlarged views of the dotted orange boxed regions are shown (bold boxes). DNA was stained with DAPI. Bars represent 10 μm. (**c**,**d**) The intensity profiles displayed correspond to the dashed blue lines in each enlarged merge view that go through a chromocenter. They show the intensity measurements in arbitrary units (a.u.) from the staining of DAPI, γH2AX and either KU70/80 (**c**) or RAD51 (**d**).

## Data Availability

Not applicable.

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
