# Peer review of "Differential Recruitment of DNA Repair Proteins KU70/80 and RAD51 upon Microbeam Irradiation with α-Particles"

_biology, 2022, doi:10.3390/biology11111652_

Round 1

Reviewer 1 Report

General Comments:  The authors have investigated the kinetics on the recruitment of proteins involved in the repair of DNA double strand breaks induced by alpha particle from MIRCOM microbeam facility using NIH-3T3 cells as experimental model.  The experimental finding demonstrated that protein (Ku70/80) involved in non-homologous end joining pathway are recruited at the site of DSB much earlier than those (RAD51) involved in homologous recombination repair pathway and concluded that the use of a microbeam such as MIRCOM, a  powerful tool to study more precisely the cellular response to ionizing irradiation at the molecular level. As facility to irradiate cell system with alpha particle in a controlled manner are limited as well the data can be used to better understand the  molecular mechanism involved in the repair of high LET irradiation induced damages.

Specific comments:

Page No-2, lines 86-97 (Introduction): Major aim of the study was to explore the molecular mechanism to investigate the repair pathway and dynamics of proteins involved or validating the MIRCOM micro-beam as option to irradiate the cells with alpha particle? Based on this aim, even the abstract require revision. Should be specific: initial few sentences gives the impression is a review article.. Then irradiation facility…few words on the recruitment of repair proteins not correlating with the title..  

 Page No-3, lines 99-110 (Cell culture): Was asynchronously growing cells were seeded for the experiments? How 10000cells/cm2 was maintained 24 hours after seeding; moreover the cells were at different phases (~50% in G1, 20% in S and 15% in G2/M); ideal would be using a synchronized culture…

 Page No-4, lines 146-158 (Irradiation): What was the average energy of individual alpha particle?  50 particle & 9 spot; total dose? How many cells were irradiated? Provide the reason for the selected condition for the irradiation? How the experiments were carried out?  Was the experiments repeated?  What was the irradiation temperature and time used for irradiation? All these are known variables alter H2Ax expression.

 Page No-6, Figure-2: Proportion of cells (G1, S, G2/M) can be represented in single bar and how it changes over time may be more easy to follow. Why the distribution of cell phase and H2AX data is not shown for the unexposed control cultures? Or if the background was subtracted then should be mention in the text or legend for better clarity.

 Page No-8, line no 305-308: …. 24 h after exposure, around 50 % of irradiated cells do not have detectable γH2AX signal 307 anymore and are thus likely to have repaired their irradiation-induced DNA damage (Figure 3). How it was attributed due to repair? H2AX phosphorylation is a kinetic even; might be faded also!!!!

 Figure-5: unable to see in the manuscript.

 Page No-11, Conclusion: Should be precise… need not have reference

 References: Require consistency;;  e.g 10 (page/ doi is missing) and 14 (capitalized title)..

Reviewer 2 Report

In their submitted manuscript, the authors describe the use of targeted charged particle microbeam irradiation to obtain data on the recruitment of different repair factors utilized in different pathways of DNA-double strand break repair. They found that the Ku heterodimer which fosters NHEJ is faster recruited compared the HR factor Rad51 after irradiation with alpha particles at the MIRCOM microbeam. In addition, they describe differences for the two repair factors depending on chromatin architecture. Whereas the authors demonstrate, that microbeam irradiation with charged particles can be successfully applied for these types of studies and provide the advantage of a relevant radiation type and controllable dose deposition compared to commonly used laser micro-irradiation, the work stays quite descriptive and provides only limited new biological insights. In this view it still might be valuable in this special issue, but more in a character of a technical application note.

The manuscript is well written, but some issues listed below should be corrected or addressed.

Unfortunately, Figure 4 seems to be missing in the manuscript. There is only the figure legend. Please add figure.

The authors connect their work to natural occurring alpha particle irradiation and health concern for the public e.g. form Radon emanating from rocks and accumulating in some houses. However, whereas under environmental conditions, only very few cells will be traversed by a single alpha particle, the authors use 50 particles per spot (for each of the nine spots) in a cell nucleus. These around 450 alpha particles (or He-ions …) deposit a non-physiologic high dose (>50Gy) might overwhelm the repair capacity and might make results difficult to relate to the environmental conditions. Mladenov et al “Strong suppression of gene conversion with increasing DNA double-strand break load delimited by 53BP1 and RAD52” NAR 2020 showed an early saturation of HR at quite low doses and a strong suppression of HR contribution at higher doses. A dose estimation (based on the geometry of the target nuclei) should be given and these aspects should be addressed in the discussion.

It is well known that DNA-PKcs and the Ku proteins are very rapidly recruited to DSBs. Whereas most kinetic studies for fast repair components were conducted using laser microirradiation, there are few using charged particles in combination with live cell imaging to determine the real time kinetics, which should be considered to be included into the discussion. e..g. Merck et al. Photobleaching setup for the biological end-station of the Darmstadt heavy-ion microprobe, NIM B 2013 or Uematsu et al. Autophosphorylation of DNA-PKCS regulates its dynamics at DNA double-strand breaks, JCB 2007.

Rad51 is a late acting component of HR known to bind (at later times) to preprocessed (resected) DSBs replacing RPA to proceed with strand invasion and described for its delayed occurrence in many papers. In this view lines 313 “Together, these data suggest that HR-mediated repair in NIH-3T3 cells occur with slower kinetics” ….&…. “In conclusion, our data suggest that NHEJ are recruitet much faster…..” imply new findings. Please rephrase e.g “suggest” by “support”

Figure 2 & 3. “*” are used for marking the statistical significance. However, it is not clear which columns were compared. Is it tested against the 5 min values? Please specify.

Even if in the end the same, from a nuclear physicist’s point of view, strictly speaking “alpha particles” describe the product of radioactive decay of larger nuclei whereas the accelerator beam consists of accelerated He2+ ions. You might consider to acknowledge the use of these termini.

Reviewer 3 Report

Dear Authors,

your research article aimed to explore the spatio- temporal- dynamics of the two main cellular pathways involved in the repair of DNA double-strand breaks, specifically induced with α-particles irradiation. KU70/80 heterodimer and RAD51 are central players for the NHEJ and HR pathways, respectively, and can be evaluate as reporters for NHEJ and HR activity in mammalian cells. The use of the MIRCOM microbeam instrument to irradiate samples at the nuclear level with micrometric precision with a predetermined number of α-particles is a useful tool to study the effect of IR at molecular level. However, I’m wondering if your results obtained in mouse NIH-3T3 cells could be validated in a human cellular model.  In alternative, the choice of this murine model instead of a human one, should be discussed.

The manuscript is generally well-written, but the Figure 4 is missing and it is not possible to evaluate this part. A part this, no other major revisions are required.

Sincerely

Round 2

Reviewer 1 Report

My original comment: Page No-6, Figure-2: Proportion of cells (G1, S, G2/M) can be represented in single bar and how it changes over time may be more easy to follow. Why the distribution of cell phase and H2AX data is not shown for the unexposed control cultures? Or if the background was subtracted then should be mention in the text or legend for better clarity.  I convinced with the answers provided by the authors.  A sentence may be mentioned in the text.

Reviewer 2 Report

just one minor correction/clarification needed:

- amount of hoechst dye used for nuclear staining: In M+M section line 120 1µM is given, whereas Fig. 1b 150 nM is written. Please make consistent.

Reviewer 3 Report

Authors were able to clarify my doubts.

I agree with the publication of the present manuscript entiteled "Differential recruitment of DNA repair proteins KU70/80 and RAD51 upon microbeam irradiation with α-particles" in Biology.
